# How Will Changes toward Pro-Environmental Behavior Play in Customers' Perceived Value of Environmental Concerns at Coffee Shops?

**Taeuk Kim** 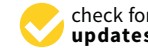 **and Sunmi Yun** *

International Center for Hospitality Research & Development, Dedman School of Hospitality,
Florida State University, Tallahassee, FL 32306, USA
* Correspondence: syun@fsu.edu; Tel.: +1-850-300-1279

**Abstract:** Our theoretical framework was designed to explain customers' decision-making process for (adopting/using?) environmentally responsible products in an eco-friendly coffee shop. We employed theory of planned behavior (TPB) and value-attitude-belief (VAB) to test their parallel mediating effect on attitudes toward environmental behavior (ATEB), perceived marketplace influence (PMI), and overall image (OI) as well as the moderating effect of switching cost (SC) on pro-environmental behavioral intentions. Data were collected through a survey of 527 customers who frequently visited a coffee shop in Korea, and structural equation modeling (SEM) was used to test the research hypotheses. The findings generally supported the hypothesized associations of the study variables within our proposed theoretical framework (ATEB, PMI, and OI of the parallel mediating effect on pro-environmental behavioral intentions) and confirmed SC's moderating effect. In addition, the study's results have important 1) theoretical and 2) practical implications for the environment. 1) This study confirmed the relationship between mediating variables on PCBI and the parallel mediating effect on PCBI as demonstrated in previous studies. 2) Furthermore, these findings might lead coffee shops to voluntarily put an end to the use of disposable products such as plastic cups or straws, which carry great environmental risk.

**Keywords:** perceived value (PV); attitude toward environmental behavior (ATEB); perceived marketplace influence (PMI); overall image (OI); switching cost (SC); pro-environmental customer behavioral intentions (PCBI); coffee shops

## 1. Introduction

Our environment is constantly changing. We need to be increasingly aware of the problems that surround it such as warming and cooling periods, different types of weather patterns, the influx of natural disasters and much more [1–4]. Furthermore, awareness of environmental consequences have become a major issue in South Korea today, and pro-environmental activity is now treated as 'essential' instead of as 'optional' [5]. The reason for South Korean's growing interest in the environment can be explained by the Chinese fine dust invasion, which has led to severe health risks [1]. Growing environmental concerns have caused Korean people to become aware of the plastic waste issue too, which was created by the Chinese refusal to import waste [6].

Within this context, the use of plastic cups in coffee shops has become a social issue. Even the Ministry of Environment began restricting the use of plastic cups in coffee shops in August 2018 [5]. These changes suggest that eco-friendly consumption is now an obligation, not a choice. Thus, it is possible to assert that environmental concerns can enhance awareness of environmental consequences, which in turn prompts pro-environmental behaviors/intentions such as visiting green coffee-shops

and making commitments [7–10]. Coffee shops in South Korea have joined this eco-friendly vibe already. Starbucks Korea introduced the use of paper straws, thereby reducing plastic straw waste on a massive scale (180 million per year) [11]. Paris Baguette made a voluntary agreement with the Ministry of Environment to cut back on the use of unnecessary disposable products and make an effort to protect the environment [12]. The fact that awareness of environmental consequences in the larger market can encourage customers to adopt pro-environmental behaviors/intentions is the reason that the coffee-shop industry is taking on sustainable management [13]. According to the Korea Consumer Agency's [14] report, 87.1% of coffee-shop customers believe "coffee shops should reduce disposable product" compared to 3.0% of customers who believe that "there's no need for Coffee-shops to ban disposable product." This reveals the increase in customers' concern with environmental responsibility and the need for eco-friendly marketing within coffee shops.

Various kinds of theoretical approaches from within the field of environmental psychology were used to predict the pro-environmental behaviors/intentions of coffee-shop customers [2,15]. The most influential approach was Ajzen's Theory of Planned Behavior (TPB) developed in 1985 [16–20]. This study used TPB as a rationale for understanding human buying behavior, grasping the stakes between synchronous factor and asynchronous factor, as they are crucial. In addition, the values and attitudes of customers who frequent eco-friendly coffee shops are essential for explaining customers' pro-environmental behavior [21]. Thus, Value-Attitude-Behavior (VAB) theory was often used to predict pro-environmental behaviors and intentions. Various studies about the value of VAB process are being made [2,22–27]. However, a crucial discovery for the relationship between TPB and VAB is still in progress. We think this study can make a large contribution to understanding customers' pro-environmental behaviors/intentions and can help determine the implications (of these behaviors?) on marketing by verifying the abovementioned theories. We also want to establish TPB using VAB and the overall image of green coffee shops. This way, we may find their importance. Even the proof of their importance in forming pro-environmental behaviors/intentions exist [7,28]. But it is also true that practical studies with VAB and green shop image together are quite rare to find. Many researchers highlight the role of transition cost in explaining pro-environmental behaviors/intentions, and for studying buying behavior and relative quality. However, only a few number of empirical studies exist [29,30].

In this study, our purpose was to develop a theoretical framework that clearly explains customers' pro-environmental behavioral intention formation in an environmentally responsible coffee shop context and to propose the academic and practical implications of the foodservice industry based on the results of the research. Specifically, we aimed to 1) verify the relationship between the construct variables, 2) test the mediating impact of study variables, 3) deepen VBN by considering the moderating impact of switching cost, and 4) broaden the TPB by incorporating the VAB framework, overall image, and switching cost.

## 2. Conceptual Framework

### 2.1. Pro-Environmental Coffee Shops

Recently, the food service industry has begun performing pro-environmental management, thereby fulfilling its social responsibility through environmental protection, saving resources, and removing environmental pollution [10,31–34]. The Green Restaurant Association (GRA) defines a pro-environmental food service company as a company that saves water and energy, decreases and recycles waste, and uses non-chemical and pro-environmental products in the long term. One of the most typical pro-environmental food service company, Starbucks, runs an eco-friendly system by using recycled take-out cups, organic foods, and reusable mugs, practices that are observable to customers. Unobservable components such as energy saving or air conditioning are also considered a part of pro-environmental management and include the use of energy-saving lights or heating and venting air conditioning systems [35]. Furthermore, Starbucks offers a "Responsibility" menu to encourage ethical

sales, protect the environment and promote local commitment. All coffee beans are also manufactured under a pro-environmental system which protects both the environment and farmers [36]. Starbucks' use of biodegradable and edible straws, as well as cups without straws, is leading pro-environmental campaigns. In this vein, pro-environmental management prefers a permanent point of view instead of a long-term view. Moreover, it pays more attention to the natural environment, focusing on all mankind instead of one society [32]. This type of management correlates with green consumerism. Pro-environmental management answers consumers' demand through communicational interaction in order to achieve a balanced society and promote human welfare [37]. Pro-environmental management of restaurants can bring energy/resource conservation, waste reduction, higher work efficiency and lower operation costs [8]. Moreover, green food can satisfy customers' need for stability and develop a competitive edge for sustainable agriculture [38]. The research of Jeong et al. [39] offers practical guidelines for effective green marketing and for creating the pro-environmental image of restaurants. Chen and Lee [40] have shown a positive relationship between the green image of Taiwanese Starbucks and its customers' purchase intention. Jang, Chung, and Kim [41] have verified a positive relationship between restaurant customers' environmental concerns and pro-environmental restaurants in Seoul, Gyeonggi province. Filimonau et al. [32] set a camera at coffee shops to examine customers' garbage dumping behavior.

## 2.2. Theory of Planning Behavior and Value-Attitude-Behavior

Theory of Planned Behavior (TPB) is a result of Ajzen's [20] attempt to modify Theory of Reasoned Action (TRA) by adding a behavioral intention variable between attitude and action. Because it is really hard to predict or control individuals' actions, as there are uncertain variables beyond one's free will, Ajzen [20] proposed TPB by considering such asynchronous variables. Since the TPB has been released and its validity/utility proven, various studies use TPB to predict behavioral intentions and actions quickly [6,12,16,42–44]. TPB is a theory that states that attitude toward behavior, subjective norm, and perceived behavioral control intention shape an individual's behavioral intention. Formed behavioral intention and perceived behavioral control then affects one's actions [20,45]. In TPB, control belief is made under the presence of factors that facilitate or hinder the behavior such as obstacles, opportunity, or resources. Less obstacles and a higher quantity of resources/opportunities increase the perceived behavioral control [46]. For instance, high environmental concern and low hesitation about environmental protection increase perceived behavioral control. Normally, high attitude toward behavior and subjective norm increase perceived behavioral control and behavioral intention [19]. In addition, the level of subjective norm can be determined by the level of an individual's normative belief. It can also be determined by level of social pressure and people's belief in whether or not they should follow such behavior based on specific individuals or contexts [47]. Garbage dumping may relate more to subjective norms while purchasing decisions may relate more to attitude toward behavior [18]. Thus, perceived behavioral control is an uncontrollable factor as it is [based on?] individuals' subjective evaluation of difficulty. It can be divided into self-efficacy, an internal control factor, and money or resources, the external control factor [45,48]. Hansen, Saridakis, and Benson's study on TPB and customers' behavior [49], added perceived risks to TPB and verified meaningful influence with behavioral intention. Leary et al. [12] showed market behaviors' mutual relation between individuals and extended previous studies about the relationship between values, beliefs, and behaviors. Chen and Tung [16] predicted customers' intention to visit green hotels and developed extended TPB which included customers' moral imperative. Han [6] explained travelers' pro-environmental behavior concerning green hotels by using Value-Belief-Norm theory and TPB. Tuwanku et al. [17] performed an experiment on customers who were aware of Starbucks' social responsibility campaign. They verified the relationship between TPB and behavioral intention, showing that all sub-variables except subjective norm had a significant influence on behavioral intention. Therefore, in this study, the following hypothesis was set up for coffee shop customers.

As an environmental behavior theory, Value-Belief-Norm (VBN) provides speculative reason for the effect of values, beliefs, and norms on pro-environmental behavior [27]. VBN theory was used as a conceptual structure by Stern to examine individuals' pro-environmental behavior/intention and social responsibility. The core three elements used to explain one's pro-environmental behavior were value, belief, and norm. An early model of VBN was organized by Stern at 1999 using past studies such as Schwartzs' value theory [50], Schwartzs' norm-activation theory [51], and Dunlap, Van Liere, and Dunlap's new environmental paradigm [52]. Environmental behavior theory has grown to include Knowledge-Attitude-Practice (KAP), the Theory of Reasoned Action (TRA), the Theory of Planned Behavior (TPB), Model of Goal directed Behavior (MGB), Responsible Environmental Behavior (REB), and Norm-Activation-Model (NAM). Such studies have been used to shed light on the principle of human behavior. Fishben and Ajzen [53] insisted that social, subjective norms and emotion norms or norms activated by marginal men can affect one's behavior in specific conditions. Stern modified some terms of the research model proposed by Stern et al. [27] and established VBN theory. Stern also provided three factors that can affect environment: egoist value, altruistic value, and ecologic value. Thereby, Stern adduced the multidimensionality of value. In addition, Stern highlighted ascription of responsibility and awareness of consequences as critical variables to predict individuals' pro-environmental behavior as they are an index of our perception of various human influences [27]. In a study related to the hospitality industry, Kiatkawsin et al. [23] found a meaningful relationship between VBN and Vroom's expectancy theory and pro-environmental behavior/intentions. Han et al. [26] managed to relate VBN and pro-environmental behavior/intentions in cruise business. Landon et al. [24] established a correlation between pro-environmental behavior/intentions and psychological mechanism, targeting a panel of U.S. Travelers with an annual household income above $50,000 and aged above 18. Han [6] revealed guests' intentions of using green hotels using a combined theory of TPB and VBN. Lee and Jan [25] applied ecologic value on nature-based tourism to find differences between normal responsible environmental behavior and geospatial responsible environmental behavior. Based on previous studies, the following hypotheses were set.

**H1.** *Perceived value has a significant effect on attitude toward environmental behavior.*

**H2.** *Perceived value has a significant effect on perceived marketplace influence.*

**H3.** *Attitude toward environmental behavior has a significant effect on pro-environmental customer behavior.*

**H4.** *Perceived marketplace influence has a significant effect on pro-environmental customer behavior.*

### 2.3. Multiple Mediation of Attitude Environmental, Perceived Marketplace Influence, and Overall Image

2.3.1. Attitude toward Environmental

According to Ajzen and Fishbein [54], attitude has been used in quasi-experimental research as an explanatory device for reading changes in human behavior under external stimuli or control measures. It has multidimensional attributes including cognitive (knowledge), affective, and conative (intentions), which are interrelated to each other [55]. However, as human behavior is complicated and unpredictable, it is not appropriate to use attitude as a single element to forecast the course of human behaviors [54]. Furthermore, recent studies have shown that relationships between attitudes and behavior have no significant feature[s] compared to general-specific issue behavior relationships [56]. Thus, McGuire [57] found the association between emotion (environmental concern) and conation (verbal intentions) in attitudinal construct by examining environmental attitudes among college students. Hartig and Kaiser [58] also showed that environmental attitudes could partially mediate the positive association between the use of natural environment for psychological restoration and higher levels of pro-environmental behaviors.

### 2.3.2. Perceived Marketplace Influence

This study connects the individual consumer to marketplace actors by exploring the role of perceived marketplace influence (PMI) in motivating ethical action. PMI is a multidimensional construct defined as "the belief that one's efforts in the marketplace can influence the marketplace behavior of other consumers and organizations" [13]. It allows consumers to utilize their marketplace-focused behaviors (e.g., word of mouth, purchase behavior) as an avenue for influencing, both, other consumers (PMI Consumer) and organizations (PMI Organization). Consumers are likely to act on pro-environmental intentions only when it seems to ensure their desired outcomes. Leary et al. [13] and Leary, Vann, and Mittelstaedt [59] applied to make the concept of PMI, maintaining that one's belief of influence on other marketplace actors (e.g., other consumers and organizations) serves an important role in motivating instrumental marketplace behavior.

### 2.3.3. Overall Image

The concept of overall image (OI) has drawn meaningful attention from both academia and industry, as it is assumed to play a critical role in the decision making process of customers [60]. People's behaviors are more likely to be determined by an image than by objective reality [61]. Thus, customers using hotels with OI have shown a significant amount of probability with regards to engaging in eco-friendly practices [39]. Similarly, restaurants with OI can persuade consumers to perceive they have pro-environmental customer behavioral intentions. In consumer behavior research, Oliver [62] found that consumer attitude toward a product is already established through prior or current information related to the product (or provider) before actual consumption/use. In other words, customers often increase their biases for or against the provider based on its image in the marketplace. Oliver mentioned these customers' attitudes, generated through these cognitive processes, as contributors in customer intention building. Based on previous studies, the following hypotheses were set.

**H5.** *Attitude toward environmental behavior has a significant effect on overall image.*

**H6.** *Perceived marketplace influence has a significant effect on overall image.*

**H7a.** *Overall image has a significant effect on word of mouth behavior.*

**H7b.** *Overall image has a significant effect on willing to pay behavior.*

**H7c.** *Overall image has a significant effect on sacrifice behavior.*

**H8a.** *Attitude toward environmental behavior, perceived marketplace influence, and overall image will be parallel mediation the relationship between perceived value and word of mouth behavior.*

**H8b.** *Attitude toward environmental behavior, perceived marketplace influence, and overall image will be parallel mediation the relationship between perceived value and willing to pay behavior.*

**H8c.** *Attitude toward environmental behavior, perceived marketplace influence, and overall image will be parallel mediation the relationship between perceived value and sacrifice behavior.*

### 2.4. Switching Cost as Moderator

Switching barriers create restrictions such as required resources or opportunity cost to perform switching after purchasing product or service use [63]. One of the most representative factors is switching cost. It is customers' awareness of cost which leads to finding alternative service providers after meeting with the current service provider [64]. The concept of switching barriers was based on the relationship between purchaser and seller in marketing business, which was developed to (reveal/show/demonstrate?) the theoretical model of service provider, customer and distribution channel. Porter [64] defined switching cost as an "incurred cost to change service provider."

Sharma and Patterson [65] also pointed out that psychological and emotional price is in switching cost. It works as a barrier when customer tries to switch, which contains relational solidarity, rapport or mutual trust between customer and provider in the course of a transaction [66]. It also contains financial cost and psychological cost for facing uncertainty [67]. As such, the relatively high cost of switching interrupts customers' switching, which may compromise customer loyalty [68]. Existing research such as Keaveney's [69] study considered Service Provider Switching Model (SPSM), which includes eight types of switching cost: price, inconvenience, core service failure, service failure in contact, recovery failure, competition, ethical problem, and involuntarily switching. Jones et al. [70] insisted that switching cost was a barrier for customer defection and a crucial factor for influencing customer attraction among service corporations. Studies about switching cost of hospitality industries, such as Han and Ryu's [29], proved a correlation between green hotel revisit intentions and switching cost. Kim [71] confirmed the meaningful association between choice factors, switching cost (intermediator), and repurchase intention in the coffee shop industry. El-Manstrly [72] found switching cost to be a regulator of the interrelationship between customer perceived value and customer loyalty in the restaurant business. Therefore, in this study, the following hypotheses were set up.

**H9a.** *Switching cost plays a significant moderating role in the relationship attitude toward environmental behavior and on word of mouth behavior.*

**H9b.** *Switching cost plays a significant moderating role in the relationship attitude toward environmental behavior and on willing to pay behavior.*

**H9c.** *Switching cost plays a significant moderating role in the relationship attitude toward environmental behavior and on sacrifice behavior.*

**H10a.** *Switching cost plays a significant moderating role in the relationship perceived marketplace influence and on word of mouth behavior.*

**H10b.** *Switching cost plays a significant moderating role in the relationship perceived marketplace influence and on willing to pay behavior.*

**H10c.** *Switching cost plays a significant moderating role in the relationship perceived marketplace influence and on sacrifice behavior.*

## 3. Materials and Methods

### 3.1. Measures and Questionnaire Development

In order to see how pro-environmental behavioral intention varies according to the perceived value of coffee shop customers' eco-friendly attitude, this study has taken measurement variables from existing research literature. However, the existing measurement literature was cruise [2], gender [73], general consumer [13,28,74], hotel and restaurant [75], religiosity [76], traveler [7,19,23] and so on; nevertheless, it was different from the coffee shop. Therefore, it was judged that there was a gap in using the measurement items in existing literature in order to know the degree of customer behavior in the coffee shop. Therefore, in order to confirm the pro-environmental behavior of coffee shop customers, this study developed a questionnaire that was used in previous research through in-depth interviews with customers who have actually visited pro-environmental coffee shops and managers of pro-environmental coffee shops. In detail, first, we selected 10 customers of the consumer group and 5 managers of the expert groups who were willing to participate in in-depth interviews. Second, I explained the purpose of the research and interview to the selected people in detail. Third, through interviews, it was recorded in detail the questionnaire items used in previous research to see if they fit the pro-environmental coffee shop. Fourth, questionnaires were modified to use in pro-environmental coffee shops based on the recorded interviews. We also conducted a pre-test (a modified version of the questionnaire) on 20 customers in a pro-environmental coffee shop which was located in downtown

Seoul (Gang-nam). Finally, to improve the clarity of the questionnaire based on the responses of the pre-test, several questionnaire items were revised and the final questionnaire was completed.

In the questionnaire, the perceived value was adopted from Han et al. [2] and was assessed with three items (e.g., "pro-environmental coffee shops offered good value for the price"). The attitude toward environmental behavior was adopted from McCray and Shrum [4], Laroche et al. [3], Han [7], and Han et al. [28]. Specifically, four items were consecutively used to measure attitude toward environmental behavior (e.g. "Recycling (e.g. coffee cup) of coffee shops is important to save natural resources"). The perceived marketplace influence was adopted from Leary et al. [13] and was assessed with three items. (e.g. "The choices I make can influence what coffee shops make and sell in the marketplace."). The overall image was adopted from Han et al. [28] and three items were employed to evaluate overall image (e.g., "Overall, I have a good image of a pro-environmental coffee shop to visit"). The switching cost was adopted from Burnham et al. [77], Han and Ryu [29], and Yang and Peterson [78] with three items (e.g., "If I switch to a pro-environmental coffee shop, I will not be able to use some services and benefits from general coffee shops, such as coupons, gift certificates, and membership services"). The pro-environmental behavior intention was adapted from Han et al. [2] and it was divided into three factors: three items of word of mouth behavior (e.g., "I will say positive things about an environmentally responsible coffee shop"), three items of willing to pay behavior (e.g., "I am willing to visit coffee shop with an environmentally responsible coffee shop when deciding on visit coffee shops in the future"), and two items of sacrifice behavior (e.g., "To protect the environment, I would be willing to accept any inconvenience in an environmentally responsible coffee shop"). The Appendices A and B shown all used measurement items that used a seven-point Likert scale from (1) "Extremely disagree" to (7) "Extremely agree" in this research. In sum, this study was used 25 items for the assessment of 7 variables.

*3.2. Data Collection Process and Data Analysis*

In order to collect the data, we used an online base survey between April 22nd and April 26th, 2019. The pro-environmental coffee shop customers were randomly surveyed using the database of a professional online panel survey company (Embrain) that considered factors such as gender, age, education level, household income level, and geographical dispersion. The total number of panelists from Embrain is about 7 million who are generally invited by South Korean to be part of the company's survey panel. Respondents first answered a screening question to indicate whether or not they had visited eco-friendly coffee shops within the past month. A total of 550 participants completed the survey (and?) the data collection process. After the elimination of inappropriate responses, 527 (96%) useable data remained. The collected data were used to evaluate the adequacy of the theoretical framework and to test the hypotheses relationships.

For the data analysis of this study, SPSS 20.0 program was used for demographic characteristics and correlation analysis while AMOS 24.0 program was used for the confirmatory factor analysis (CFA), structural equation modeling (SEM), parallel mediating effect, and moderating effect.

## 4. Result

*4.1. Characteristic of Respondents*

Of the 527 respondents, 216 respondents (41%) were female and 311 respondents (59%) were male. Among the participants, 41.4% were 20–29 years old (218 respondents), 38.1% were 30–39 years old (201 respondents), 12.9% were 40–49 years old (68 respondents), and 7.6% were 50 years old or over (40 respondents). Present marital status was highest for single people (291 respondents, 55.8%), followed by married people (200 respondents, 38%), divorced individuals (21 respondents, 4%), widowed individuals (5 respondents, 0.9%), and those who were separated (7 respondents, 1.3%). The education level was generally high. 231 respondents (43.8%) were 2-year college graduates, 176 respondents (33.4%) were 4-year university graduates, 97 respondents (18.4%) were high school graduates and below, and 23 respondents (4.4%) had attended graduate school or more. The annual household income of $20,000–$39,999 was the highest, followed by under $20,000 (130 respondents, 24.7%), $40,000–$59,999 (124 respondents, 23.5%), $60,000–$79,999 (42 respondents, 8.0%), $100,000$

or over (27 respondents, 3.4%), and $80,000–$99,999 (18 respondents, 3.4%). Among the participants, approximately 234 respondents (4.44%) visited environmental coffee shops 5–8 times within a month, followed by 209 respondents (39.7%) who visited 9–12 times, 70 respondents (13.3%) who visited 13–16 times, 11 respondents (2.1%) who visited 1–4 times, and 3 respondents (0.6%) who visited 17 times or more. The purpose of visiting coffee shops was to meet friends (149 respondents, 28.3%), study (145 respondents, 27.5%), attend a business meeting (123 respondents, 23.3%), enjoy private time (60 respondents, 11.4%) and for other reasons (47 respondents, 9.5%) in that order.

## 4.2. Common Method Bias

In this study, procedural remedies and statistic remedies proposed by Podsakoff, MacKenzie, Lee, and Podakoff [79] were used to solve the problem of Common Method Bias (CMB). First, the procedural remedies were used to minimize the occurrence of CMB in the research design stage. (1) Each scale was measured separately using a different cover story. (2) Anonymity was guaranteed, which reduced the anxiety that a disadvantage would occur due to the response. (3) A counterbalance question order was presented to the respondents in order to diminish psychological separation. (4) The ambiguous measurement items were improved and used.

In addition, statistic remedies were taken to ensure that the procedural remedies were effective and to control CMB completely. In this study, Harman's single-factor test was used which is the most widely-used statistical method. As in Appendix B, the model fit Harman's single-factor model in which all observational variables were loaded on a single factor. It appeared to be highly inadequate. On the other hand, the model fit of the original measurement model was excellent. In addition, the explanatory power (see squared multiple correlations) of each observation variable of the single factor accounted for an average of 65.2%, from a minimum of 43.2% to a maximum of 85.1%. In contrast, in the 8-factor measurement model, the explanatory power of the latent variables for observational variables ranged from a minimum of 46.6% to a maximum of 91.2%, which accounted for an average of 80.8%. Thus, a single factor did not account for all of the variances of the 25-item measurement instrument. As a result, Harman's single-factor test was unlikely to have a significant impact on CMB results.

## 4.3. Confirmatory Factor Analysis

Table 1 summarizes the result of Confirmatory Factor Analysis (CFA) and the correlation analysis that provided an excellent fit to the data (Goodness-of-fit statistics for the measurement model: $\chi^2 = 447.682$, df = 247, $p < 0.000$, $\chi^2$/df = 1.812, RMSEA = 0.039, CFI = 0.978, IFI = 0.979, TLI = 0.974). The multi-item measures for each construct significantly conducted all factor loadings. All values for composite reliability were shown within a range between 0.652 and 0.912 (as shown in Table 1). The factor loadings were much higher than the level of 0.600 suggested by Bagozzi and Yi [80]. Thus, the composite reliability was well explained. Also, the correlation and Average Variance-Extracted (AVE) values were evaluated the convergent validity and discriminant validity of all the measures used AVE estimates were constructed ranged from 0.628 to 0.759 (as shown in Table 1). These values were all greater than that level of 0.500 suggested by Fornell and Lacker [81] and were higher than the square of correlations between a pair of variables. Thus, the convergent validity and discriminant validity in the study model was better explained. In addition, the construct reliability (CR) for 8 study constructs was calculated. The values of CR ranged from 0.771 to 0.904 (as shown in Table 1). These values highly exceeded significant values. Bagozzi and Yi's [80] suggested cutoff of 0.700, and this evidence supported convergent validity.

**Table 1.** The measurement model and correlation.

| Construct | Mean (SD) | AVE (CE) | PV | ATEB | PMI | SC | OI | WOMB | WIPB | SB |
|---|---|---|---|---|---|---|---|---|---|---|
| PV | 3.899 (1.171) | 0.666 (0.850) | 1 | - | - | - | - | - | - | - |
| ATEB | 4.066 (0.879) | 0.657 (0.884) | 0.429 *** | 1 | - | - | - | - | - | - |
| PMI | 3.902 (0.993) | 0.637 (0.840) | 0.545 *** | 0.596 *** | 1 | - | - | - | - | - |
| SC | 3.985 (0.933) | 0.701 (0.903) | 0.339 *** | 0.507 *** | 0.532 *** | 1 | - | - | - | - |
| OI | 4.053 (1.012) | 0.667 (0.855) | 0.473 *** | 0.600 *** | 0.678 *** | 0.491 *** | 1 | - | - | - |
| WOMB | 4.103 (1.038) | 0.759 (0.904) | 0.520 *** | 0.596 *** | 0.875 *** | 0.544 *** | 0.721 *** | 1 | - | - |
| WIPB | 4.155 (1.046) | 0.732 (0.891) | 0.510 *** | 0.600 *** | 0.826 *** | 0.538 *** | 0.722 *** | 0.919 *** | 1 | - |
| SB | 3.966 (1.147) | 0.628 (0.771) | 0.530 *** | 0.562 *** | 0.703 *** | 0.540 *** | 0.602 *** | 0.778 *** | 0.805 *** | 1 |

Note 1. PV = perceived value, ATEB = attitude toward environmental behavior, PMI = perceived marketplace; influence, OI = overall image, SC = switching cost, WOMB = word of mouth behavior, WTPB = willing to pay behavior, SB = Sacrifice behavior; SD = standardized deviation, AVE = average variance extracted, CR = composite reliability; Note 2. Goodness-of-fit statistics for the measurement model: $\chi^2$ = 447.682, df = 247, $p < 0.000$, $\chi^2$/df = 1.812, RMSEA = 0.039, CFI = 0.978, IFI = 0.979, TLI = 0.974, *** $p < 0.001$; Note 3. All factors loadings are significant at $p < 0.001$ Bold figures represent first-order factor loadings.

### 4.4. Structural Equation Modeling

Structural Equation Modeling (SEM) was conducted to test the hypothesized relationship between the variables empirically. The study model consisted of internal variables (perceived value), external variables (word of mouth behavior, willing to pay behavior, sacrifice behavior), and mediation variable (attitude toward environmental behavior, perceived marketplace influence, and overall image), and the SEM analysis was performed. For the study, a covariance matrix and maximum likelihood estimation were used. The result of SEM revealed an excellent model fit of the data (Goodness-of-fit statistics for the structural model: $\chi^2$ = 519.550, df = 176, $p < 0.001$, $\chi^2$/df = 2.952, RMSEA=0.061, CFI = 0.955, IFI = 0.955, TLI = 0.946). In addition, perceived value gave prediction power for attitude toward environmental behavior ($R^2$ = 0.223), perceived marketplace influence ($R^2$ = 0.346), overall image ($R^2$ = 0.532), word of mouth behavior ($R^2$ = 0.848), willing to pay behavior ($R^2$ = 0.898), and sacrifice behavior ($R^2$ = 0.637). The result of this study is summarized in Table 2 and Figure 1.

The path estimates show that perceived value had a significantly positive direct effect on the attitude toward environmental behavior ($\beta = 0.472$, $p < 0.001$) and perceived marketplace influence ($\beta = 0.588$, $p < 0.001$). Thus, H1 and H2 were supported. The result of estimation indicated that attitude toward environmental behavior had a significant positive effected on word of mouth behavior ($\beta = 0.116$, $p < 0.001$), willing to pay behavior ($\beta = 0.135$, $p < 0.001$), and sacrifice behavior ($\beta = 0.209$, $p < 0.001$). Thus, H3a, H3b, and H3c were supported. The impact of the perceived marketplace influence on word of mouth behavior ($\beta = 0.860$, $p < 0.001$), willing to pay behavior ($\beta = 0.792$, $p < 0.001$), and sacrifice behavior ($\beta = 0.712$, $p < 0.001$) was assessed, thereby supporting H4a, H4b, and H4c. The proposed impact of attitude toward environmental behavior and perceived marketplace influence on overall image was assessed. As expected, the attitude toward environmental behavior and perceived marketplace influence had an impact on overall image ($\beta = 0.327$, $p < 0.01$, $\beta = 0.568$, $p< 0.01$). As such, H5 and H6 were supported. The result of estimation indicated that overall image had a significant positive (effect?) on willing to pay behavior ($\beta = 0.105$, $p < 0.01$) and had an insignificant impact on word of mouth behavior and sacrifice behavior. Therefore, only H7b was supported.

Moreover, the results of the analysis of parallel mediation effect are shown via several significant indirect effects in Table 3. Results revealed that perceived value significantly affected word of mouth behavior ($\beta = 0.562$, $p < 0.01$), willing to pay behavior ($\beta = 0.555$, $p <0.01$), and sacrifice behavior ($\beta = 0.495$, $p < 0.01$) indirectly through attitude toward environmental behavior, perceived marketplace

influence, and overall image. This confirmed that all of them are all in partial mediation. Thus, H8a, H8b, and H8c were all supported.

**Table 2.** The structural model results and hypotheses testing.

| Hypothesized Paths | Coefficients | t-Values |
|---|---|---|
| H1: PV → ATEB | 0.472 | 9.082 *** |
| H2: PV → PMI | 0.588 | 11.524 *** |
| H3a: ATEP → WOMB | 0.116 | 3.458 *** |
| H3b: ATEP → WTPB | 0.135 | 3.799 *** |
| H3c: ATEP → SB | 0.209 | 4.197 *** |
| H4a: PMI → WOMB | 0.860 | 16.500 *** |
| H4b: PMI → WTPB | 0.792 | 15.526 *** |
| H4c: PMI → SB | 0.712 | 10.745 *** |
| H5: ATEP → OI | 0.327 | 6.024 *** |
| H6: PMI → OI | 0.568 | 7.641 *** |
| H7a: OI → WOMB | 0.068 | 1.425 |
| H7b: OI → WTPB | 0.105 | 2.075* |
| H7c: OI → SB | 0.004 | 0.061 |
| | **Indirect effect:** | **Total effect:** |
| H8a: PV → ATEB & PMI & OI → WOMB | 0.562 ** | 0.519 ** |
| H8b: PV → ATEB & PMI & OI → WTPB | 0.555 ** | 0.584 ** |
| H8c: PV → ATEB & PMI & OI → SB | 0.495 ** | 0.594 ** |

| Explained variable: | $R^2$(ATEP) = 0.223 | $R^2$(OI) = 0.532 | $R^2$(WTPB) = 0.898 |
|---|---|---|---|
| | $R^2$(PMI) = 0.346 | $R^2$(WOMB) = 0.848 | $R^2$(SB) = 0.637 |

Note 1. PV = perceived value, ATEB = attitude toward environmental behavior, PMI = perceived marketplace influence, OI = overall image, SC = switching cost, WOMB = word of mouth behavior, WTPB = willing to pay behavior, SB = sacrifice behavior; Note 2. Goodness-of-fit statistics for the structural model: $\chi^2$ = 519.550, df =176, $p < 0.001$, $\chi^2$/df = 2.952, RMSEA = 0.061, CFI = 0.955, IFI = 0.955, TLI = 0.946, * $p < 0.05$, ** $p < 0.01$, *** $p < 0.001$.

### 4.5. The Moderating Effect of the Switching Cost

To assess the proposed moderating impact of switching cost, a test for a metric variance was conducted. In this case, continuous variable such as an interval scale or ratio scale was used as a moderated variable to evaluate metric variable. The sample was divided into high (n = 230) and low (n = 297) switching cost groups by the K-mean cluster analysis. The baseline model showed acceptable level for the data suitability (Goodness-of-fit statistics for the baseline model: $\chi^2$ = 843.294, df = 352. $p < 0.001$, $\chi^2$/df = 2.396, RMSEA = 0.05, CFI = 0.921. IFI = 0.922, TLI = 0.908). Table 3 and Fig.1 exhibited the details related to the invariance test. This model was then compared to a series of nested models for the assessment of the hypothesized moderating role of switching cost. Hypotheses 9a, 9b, and 9c were tested. Findings indicated that the paths from attitude toward environmental behavior—word of mouth behavior ($\Delta\chi^2$ [1] = 10.672, $p < 0.05$), from attitude toward environmental behavior—willing to pay behavior ($\Delta\chi^2$ [1] = 9.522, $p < 0.05$), and from attitude toward environmental behavior—sacrifice behavior ($\Delta\chi^2$ [1] = 11.265, $p < 0.05$) included a significant difference between the high and low switching cost groups. This supported H9a, H9b, and H9c.

Next, we tested the impact of switching cost on the relationship between perceived marketplace influence and pro-environmental behavior (word of mouth behavior, willing to pay behavior, and sacrifice behavior). Results revealed that the link between perceived marketplace influence to word of mouth behavior ($\Delta\chi^2$ [1] = 10.809, $p <.01$), willing to pay behavior ($\Delta\chi^2$ [1] =9.829, $p < 0.01$), and sacrifice behavior ($\Delta\chi^2$ [1] = 5.283, $p < 0.01$) included a significant difference between the high and low switching cost groups. This supported H10a, H10b, and H10c.

Overall, these results provided empirical evidence that switching cost moderated the relationships between attitude toward environmental behavior and pro-environmental behavior and between perceived marketplace influence and pro-environmental behavior.

**Table 3.** The result of the moderating effect.

| | High SC Group (n = 230) | | Low SC Group (n = 297) | |
|---|---|---|---|---|
| | Coefficients | t-Values | Coefficients | t-Values |
| H9a: ATEP—WOMB | 0.107 | 2.620 ** | 0.101 | 2620 ** |
| H9b: ATEP—WTPB | 0.095 | 2.247 * | 0.092 | 2.247 * |
| H9c: ATEP—SB | 0.169 | 2.937 ** | 0.176 | 2.937 ** |
| H10a: PMI—WOMB | 0.883 | 15.364 *** | 0.883 | 15.364 *** |
| H10b: PMI—WTPB | 0.822 | 14.427 *** | 0.844 | 14.427 *** |
| H10c: PMI—SB | 0.659 | 9.793 *** | 0.727 | 9.797 *** |
| | Baseline model (Freely Estimated) | Nested model (Constrained to be Equal) | Chi-square difference test: | Test results: |
| H9a: ATEP—WOMB | $\chi^2$ (352) = 854.568 | $\chi^2$ (353) = 843.896 | $\Delta\chi^2$ (1) =10.672, $p < 0.05$ | Supported |
| H9b: ATEP—WTPB | $\chi^2$ (352) = 854.568 | $\chi^2$ (353) = 845.047 | $\Delta\chi^2$ (1) =9.522, $p < 0.05$ | Supported |
| H9c: ATEP—SB | $\chi^2$ (352) = 854.568 | $\chi^2$ (353) = 843,303 | $\Delta\chi^2$ (1) = 11.265, $p < 0.05$ | Supported |
| H10a: PMI—WOMB | $\chi^2$ (352) = 854.568 | $\chi^2$ (353) = 843.759 | $\Delta\chi^2$ (1) = 10.809, $p < 0.05$ | Supported |
| H10b: PMI—WTPB | $\chi^2$ (352) = 854.568 | $\chi^2$ (353) = 843.739 | $\Delta\chi^2$ (1) = 9.829, $p < 0.05$ | Supported |
| H10c: PMI—SB | $\chi^2$ (352) = 854.568 | $\chi^2$ (353) = 843.286 | $\Delta\chi^2$ (1) = 5.283, $p < 0.05$ | Supported |

Note 1. PV = perceived value, ATEP = attitude toward environmental behavior, PMI = perceived marketplace influence, OI = overall image, SC = switching cost, WOMB = word of mouth behavior, WTPB = willing to pay behavior, SB = sacrifice behavior; Note 2. Goodness-of-fit statistics for the baseline model: $\chi^2$ = 843.294, df = 352. $p < 0.001$, $\chi^2$/df = 2.396, RMSEA = 0.05, CFI = 0.921. IFI = 0.922, TLI = 0.908.

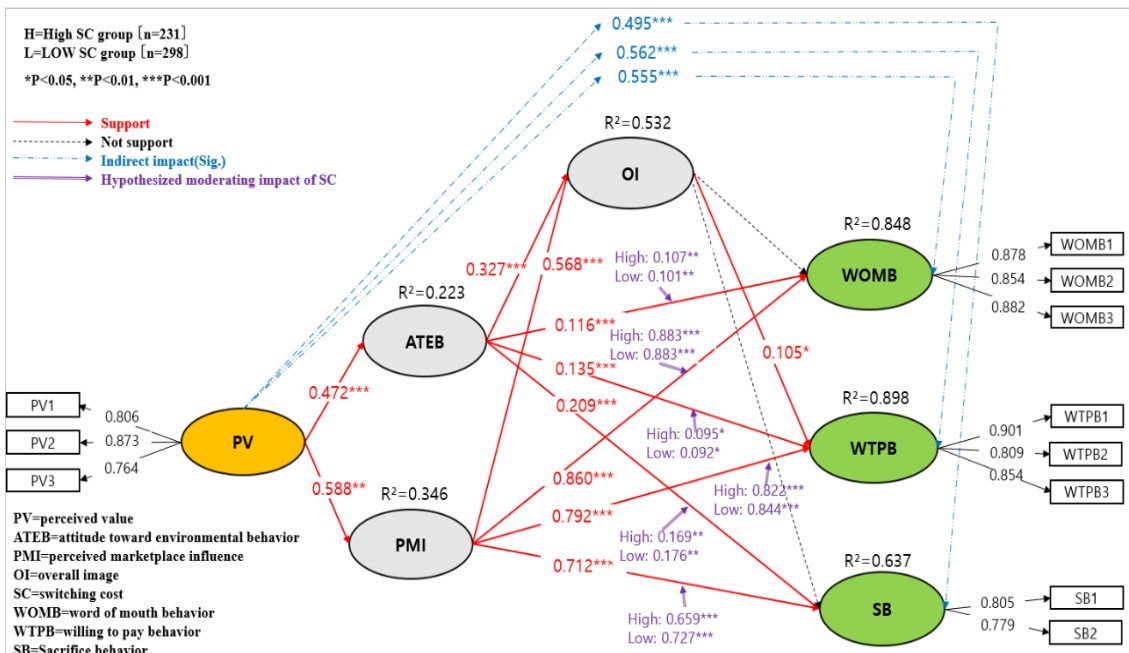

**Figure 1.** Structural equation model estimation and test for structural metric invariance.

## 5. Results and Discussion

### 5.1. Conclusion

The purpose of this study was to explain the pro-environmental behavior intention of customers who visit eco-friendly coffee shops by using the most effective Theory of Planning Behavior (TPB) and Value-Attitude-Behavior (VAB) to predict consumer behavior. Also, it tried to identify the values and attitudes that affect the pro-environmental behavior intention in this process and to verify the influential relationship between them. In the pro-environmental coffee shop context, there is limited scholarly research on behavioral intention for environmentally responsible practices during pro-environmental coffee shop visits. The proposed theoretical framework comprising coffee shop practices of perceived value as independent variables included attitude toward environmental behavior, perceived marketplace influence, and overall image as mediators; switching cost as a moderator was demonstrated to be useful and to satisfactorily predict word of mouth behavior, willing to pay behavior, and sacrifice behavior. The results of the study were as follows. (1) Perceived value had a significantly positive direct effect on the attitude toward environmental behavior and perceived marketplace influence. (2) Attitude toward environmental behavior and perceived marketplace influence had a significant positive effect on word of mouth behavior, willing to pay behavior, sacrifice behavior, and overall image. (3) Overall image only had a significant positive effect on word of mouth, willing to pay behavior. (4) Perceived value significantly affected word of mouth behavior, willing to pay behavior, and sacrifice behavior indirectly through attitude toward environmental behavior, perceived marketplace influence, and overall image, thereby confirming all of them are in partial mediation. (5) The switching costs had shown moderating effect.

### 5.2. Implications

This study has several theoretical and practical implications. First, an explanation of customers' pro-environmental behavior intentions (PCBI) based on TPB might provide a rationale for further research of coffee shop customers' PCBI. In addition, this would benefit PCBI study of restaurant and hotel industry besides coffee shop industry.

Second, a relationship between VAB model and overall image has been found, which might facilitate the establishment of extended TPB. This study stressed the importance of a green coffee shop image and instilled the implications of the VAB model, allowing it to take an important position in the study of PCBI. Third, previous studies about coffee shop customers' PCBI only focused on attitude, using it as a parameter in the VAB model to determine the perceived value and PCBI [24–26]. This study, however, used both "attitude toward environmental behavior" and "perceived marketplace influence and overall image" as the parameter and analyzed partial mediation effect. This can prove the fact that appearance of parameter between the independent variable and dependent variable may result in complex effect, affecting the outcome variable.

Third, there was no empirical study of switching cost. Previous studies referred gravity of switching cost to stress deduction of customer PCBI. In this study, switching cost is used as the moderating effect to help explain moderating effect of customer behavior based on perceived environmental customer value. Thus, switching cost gets more reason to be treated as the moderating effect in future studies.

Forth, the study has shown that the perceived value of coffee shop customers can affect both "attitude toward environmental behavior" and "perceived marketplace influence and overall image". This supports arguments made by previous studies in [6,23,25–27]. Furthermore, the study related to management of state also supports this, providing the fact that perceived environmental value among the citizen can affect attitude toward environment and its overall image [82]. As the perceived value of green coffee shops among customers becomes higher, "attitude toward environmental behavior" and "perceived marketplace influence" become more positive. It takes the same track with the general understanding of marketing territories; perceived value and attitude toward product and service have

a positive relationship [83]. Thus, a strategic approach about enhancing customers' perceived value should be found.

Fifth, the overall image of green coffee shops has no effect on world of mouth or sacrificial intention, and has only small amount of effect on purchase intention. Compared to previous studies, this is an opposite conclusion [28,61,62,75]. Perceived marketplace influence has a greater impact on customers' behavior/intentions than attitude toward environmental behavior. However, attitudes toward environmental behavior [and?] toward customers' behavior/intentions have greater deviation than perceived marketplace influences. To sum up, overall image or attitude toward environmental behavior of coffee shops has no impact on PCBI. However, perceived marketplace influence has a big impact on PCBI. We need to focus on this. The fact that only the perceived marketplace influence has an impact on PCBI can be translated to a lack of coffee shop customers' environmental awareness. This can be related to Korean culture itself. Korean people depend on the gaze of others, not one's own, to evaluate themselves [84]. This means Korean people can visit green coffee shops in the condition of other people visiting green coffee shops and care for the environment.

Sixth, attitudes toward environmental behavior, perceived marketplace influence, and overall image were found to be partially mediated by perceived value and behavioral intentions (word of mouth behavior, willing to pay behavior, and sacrifice behavior). These results are consistent with the results of Han et al. [2], Leary et al. [13], Landon et al. [24], Lee and Jan [25], Han et al. [26], McGuire [55], McGuire [57], and Lita et al. [75]. This result suggests the existence of "attitude toward environmental behavior, perceived marketplace influence, and overall images'" in building PCBI. As the perceived marketplace influence and overall image can also affect PCBI, it is not efficient to prohibit them but use the perceived value and attitude toward environmental behavior only in building customers' PCBI. Using them together or pairing them might have synergy, so to speak. Thus, green coffee shop companies should do their best to take responsibility for making a positive impact on the green coffee shop market.

Lastly, even though there was only a small difference between the group with high switching cost and low switching cost, switching cost of attitude toward environmental behavior and pro-environmental behaviors have greater impact on high groups' pro-environmental behaviors, while the switching cost of perceived marketplace influence and pro-environmental behaviors have greater impact on low groups' pro-environmental behaviors. In other words, a group with high switching costs is more likely to have attitude toward environmental behavior. In this context, groups with low switching cost are more likely to have PCBI via perceived marketplace influence. Assembling this result can draw this conclusion; switching cost can greatly affect PCBI. Thus, to enhance the PCBI of customers with perceived market influence, the solution to lowering the switching cost for green coffee shops has to be found.

## 6. Limitations and Future Studies

This study can provide a rationale for studying PCBI and takes an important position in it. However, there are certain limitations to consider. First, the number of survey samples is relatively low as the research area is limited to South Korea. Further research should be conducted in larger regions, such as the USA or Europe. To do so, cross culture studies should also be conducted, as there would be various differences between countries. Second, this study chose participants who visited green coffee shops over the course of a month. However, as the memory ability of human beings is limited and false memory can be made in long-term memory, further studies should choose samples with comparatively recent experiences; This will help minimalize the risk of false memories. Third, there are various factors that can affect PCBI. However, this study only focused on perceived value, attitude toward environmental behavior, perceived marketplace influence, and overall image. Moreover, it did not include demographic features, which can have a profound impact. Lastly, as this study focused on green customer behaviors, making further study about green company management would be desirable. Thus, a study with multi-variable factors and reflection of demographic feature is necessary.

**Author Contributions:** Conceptualization, T.K. and S.Y.; methodology, S.Y.; formal analysis, T.K. and S.Y.; investigation, T.K.; resources, S.Y.; data curation, S.Y.; writing—original draft preparation, T.K. and S.Y.; writing—review and editing, T.K. and S.Y.

**Funding:** This research received no external funding.

**Conflicts of Interest:** The authors declare no conflict of interest.

## Appendix A

**Table A1.** Measurement items.

| **Perceived value** |
| --- |
| Visiting a pro-environmental coffee shop was worth the money paid. |
| Visiting a pro-environmental coffee shop was a pleasant experience. |
| Visiting a pro-environmental coffee shop was to improve self-esteem. |
| **Attitude toward environmental behavior** |
| In our country, we have enough electricity, water, and trees that we do have to worry about consecration. |
| Recycling is too much trouble (reverse cored) |
| Recycling is important to save natural resources |
| The Coffee shops are concerned about the environment. |
| **Perceived marketplace influence** |
| I believe my individual efforts to be environmentally friendly will persuade others in my community to purchase environmentally friendly products |
| The choices I make can influence what companies make and sell in the marketplace |
| If I buy environmentally friendly products, companies will introduce more of them |
| **Switching cost** |
| If I switch to a pro-environmental coffee shop, I will not be able to use some services and benefits from general coffee shops, such as coupons, gift certificates, and membership services |
| Switching to a pro-environmental coffee shop will incur a monetary cost, such as no discounts, no special offers, and paying a higher price for coffee and dessert than general coffee shops |
| Even if I have enough information of a pro-environmental coffee shop, comparing the eco- environmental coffee shop with general coffee shops another takes a lot of energy, time, and effort |
| In general, it would be a hassle switching to pro-environment coffee shop. |
| **Overall image** |
| Overall, image for staying in a pro-environmental coffee shop is positive |
| Overall, image I have about a pro-environmental coffee shop is positive |
| Overall, I have a good image about a pro-environmental coffee shop to visit |
| **Word of mouth behavior** |
| I will encourage my friends and relatives to choose an eco-friendly coffee shop. |
| If someone is looking for a coffee shop, I will advise him/her to choose an environmentally responsible coffee shop. |
| I will say positive things about an environmentally responsible coffee shop. |
| **Willing to pay behavior** |
| I am willing to visit coffee shop with an environmentally responsible coffee shop when deciding on visit coffee shop in the future. |
| I plan to visit coffee shop by environmentally responsible coffee shop instead of a regular coffee shop in the future. |
| I will expend effort on visiting by an environmentally responsible coffee shop instead of a general coffee shop in the future. |
| **Sacrifice behavior** |
| To protect the environment, I would be willing to pay more for an environmentally responsible coffee shop. |
| To protect the environment, I would be willing to accept any inconvenience (e.g., recycling, reducing water/ energy use, decreasing waste, using the recycling coffee cup) in an environmentally responsible coffee shop. |

## Appendix B

**Table A2.** The Result of Harman's Single-Factor Test.

| Observation Variable | Harman's Single-Factor Model | 8-Factor Measurement Model |
|:---:|:---:|:---:|
| PV1 | 0.432 | 0.806 |
| PV2 | 0.523 | 0.873 |
| PV3 | 0.574 | 0.764 |
| ATEB1 | 0.481 | 0.729 |
| ATEB2 | 0.619 | 0.863 |
| ATEB3 | 0.588 | 0.836 |
| ATEB4 | 0.573 | 0.808 |
| PM11 | 0.720 | 0.791 |
| PMI2 | 0.676 | 0.759 |
| PMI3 | 0.764 | 0.843 |
| SC1 | 0.485 | 0.757 |
| SC2 | 0.573 | 0.466 |
| SC3 | 0.613 | 0.867 |
| SC4 | 0.526 | 0.854 |
| OI1 | 0.529 | 0.652 |
| OI2 | 0.660 | 0.862 |
| OI3 | 0.729 | 0.912 |
| WOMB1 | 0.844 | 0.878 |
| WOMB2 | 0.806 | 0.854 |
| WOMB3 | 0.838 | 0.882 |
| WTPB1 | 0.851 | 0.901 |
| WTPB2 | 0.772 | 0.809 |
| WTBB3 | 0.803 | 0.854 |
| SB1 | 0.673 | 0.805 |
| SB2 | 0.659 | 0.779 |
| Minimum | 43.2% | 46.6% |
| Maximum | 85.1% | 91.2% |
| Mean | 65.2% | 80.8% |

Note 1. PV = perceived value, ATEB = attitude toward environmental behavior, PMI = perceived marketplace influence, SC = switching cost, OI = overall image, WOMB = word of mouth behavior, WTPB = willing to pay behavior, SB = sacrifice behavior; Note 2. Goodness-of-fit statistics for the Harman's single-factor model: $\chi^2$ = 3015.814, df = 275, $p < 0.000$, $\chi^2$/df = 10.967, RMSEA = 0.138 CFI = 0.705, IFI = 0.706, TLI = 0.678; Note 3. Goodness-of-fit statistics for the measurement model: $\chi^2$ = 447.682, df = 247, $p < 0.000$, $\chi^2$/df = 1.812, RMSEA = 0.039, CFI = 0.978, IFI = 0.979, TLI = 0.974.

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
