# Peer review of "How Will Changes toward Pro-Environmental Behavior Play in Customers’ Perceived Value of Environmental Concerns at Coffee Shops?"

_sustainability, doi:10.3390/su11143816_

Reviewer 1 Report

1. The structure of the paper needs attention and the usual rule (introduction-rationale-need for the work/research questions, background-literature review, approach-methods-research performed, results, discussion and then conclusions/concluding remarks) should be followed more closely to facilitate the flow of the paper. Please develop further your discussion by drawing on relevant studies and in relation with prior MDPI's Sustainability Journal literature - develop further and expand your final section of concluding remarks; incorporate research and policy recommendations in the final conclusion section. Cite (primarily) in these final-most critical sections of your manuscript relevant papers published in the Journal you submitted your work to (in order to provide some sort of continuity of the specific research string).

2. More references to recent & relevant literature/empirical studies could increase the quality of the research paper and provide a much clearer message to the reader - these may help you building your discussion which needs to be extended (including the scope of organizational sustainability and perhaps national culture); add the following to your reference list:

Halkos G... (2016). National CSR and institutional conditions: An exploratory study. Journal of Cleaner Production, 139, 1150-1156.

Halkos G... (2017). Revisiting the relationship between corporate social responsibility and national culture: A quantitative assessment. Management decision, 55(3), 595-613.

Skouloudis A... (2015). Priorities and perceptions for corporate social responsibility: an NGO perspective. Corporate Social Responsibility and Environmental Management, 22(2), 95-112.

Skouloudis A... (2015). Priorities and perceptions of corporate social responsibility: Insights from the perspective of Greek business professionals. Management Decision, 53(2), 375-401.

3. The introductory/opening section should communicate a little clearer the literature gaps, as weel as the study's aims & objectives in order to facilitate the flow of the study. The title needs refinement; too long. The abstract needs rewriting; too many abbrevations which (in general) should be avoided both in titles and abstracts.

4. Concluding remarks - authors must elaborate more on what is their contribution to the literature as well as on opportunities for future research. Further emphasis must be placed on the following: Why such assessment is important? and how it extend so existing knowledge on the issue/topic? Conclusions need to be written in a clear and coherent manner and draw the main lessons from the paper. I suggest you to concentrate on the description of the implications of the work, the main findings and its potential replicability - empirical investigation. Furthermore, limitations of the study need to be outlined to a greater extent, and so are any potential connections between your study and specific aspects of the Journal's scope.

5. Carefully check the references, so as to make sure they are all complete and follow the Guidelines to Authors.

6. Finally, please do check thoroughly, in order to avoid grammar, syntax or structure/presentation flaws (e.g. some figures need refinement). Make sure you retain a formal/academic-specific style of presenting your work throughout the text - (if necessary) please seek for professional English proofreading services or ask a native English-speaking colleague of yours in order to refine and improve the English in your paper.

Author Response

Dear  reviewer,

Thank you very much for reviewing our manuscript. We appreciate the opportunity you have afforded us to revise and resubmit. We found your suggestions to be thought-provoking and useful and have worked diligently to improve the manuscript as you suggested. Below, you will find our replies and responses to your constructive comments. Within the responses, red sections denote changes we made to the manuscript itself. Within the manuscript itself, changes are highlighted in red. We hope the changes made are satisfactory to you.

Reviewer 2 Report

The submitted manuscript deals with the topic of sustainable consumption. The manuscript aim is to predict the pro-environmental behaviour intention of consumers who visited eco-friendly cofee shop by integrating the Theory of Planner Behaviour (TPB) and the Value-Attitude-Behaviour (VAB).

The manuscript is interesting and timely and it faces a topic which has a rather high importance in the consumers studies literature. The manuscript is well-structured and clearly presented. The conceptual framework is divided in four subsection to clearly present the contribution of the literature in enahncing consumer knowledge. From my point of view, the manuscript could be suitable to be pubblished on Sustainability. However, there are few points that need to be addressed in the revision phase to improve readers' understanding.

More in depth, (page 3- line 102) you mentioned that this type of management correlates with consumerism. Please, try to better explain the concept of consumerism, since in the literature could be understanding "green consumerism", "ethical consumerism", or even "mass consumerism". You can help readers understanding by adding a brief statement in which you explain the term consumerism.

I suggest renaming "results" section in "results and discussion"

Moreover, page 12 line 443: also in this case, please, rename this section in "Conclusions, limitations, implication and further research.

Hope my suggestions are useful for the authors.

Author Response

Dear  reviewer,

Thank you very much for reviewing our manuscript. We appreciate the opportunity you have afforded us to revise and resubmit. We found your suggestions to be thought-provoking and useful and have worked diligently to improve the manuscript as you suggested. Below, you will find our replies and responses to your constructive comments. Within the responses, red sections denote changes we made to the manuscript itself. Within the manuscript itself, changes are highlighted in red. We hope the changes made are satisfactory to you.

Round  2

Reviewer 1 Report

Accept